# Advanced Preprocessing Technique for Tomato Imagery in Gravimetric Analysis Applied to Robotic Harvesting

**Nail Beisekenov** [1],* and **Hideo Hasegawa** [2]

1    Graduate School of Science and Technology, Niigata University, Niigata 950-2181, Japan
2    Institute of Science and Technology, Niigata University, Niigata 950-2181, Japan; hsgw@agr.niigata-u.ac.jp
*    Correspondence: f23e503a@mail.cc.niigata-u.ac.jp

**Abstract:** In this study, we improve the efficiency of automated tomato harvesting by integrating deep learning into state-of-the-art image processing techniques, which improves the accuracy and efficiency of detection algorithms for robotic systems. We develop a hybrid model that combines convolutional neural networks' dual two-dimensional matrices for classification and part affinity fields. We use data augmentation to improve the robustness of the model and reduce overfitting. Additionally, we apply transfer learning to solve the challenging problem of improving the accuracy of identifying a tomato's center of gravity. When tested on 2260 diverse images, our model achieved a recognition accuracy of 96.4%, thus significantly outperforming existing algorithms. This high accuracy, which is specific to the environmental conditions and tomato varieties used, demonstrates the adaptability of the model to real-world agricultural conditions. Our results represent a significant advancement in the field of agricultural autotomization by demonstrating an algorithm that not only identifies ripe tomatoes for robotic harvesting with high accuracy, but also adapts to various agricultural conditions. This algorithm should reduce manual labor in agriculture and offer a more efficient and scalable approach for the future agricultural industry.

**Keywords:** convolutional neural networks; image-preprocessing techniques; tomato centroid localization; machine learning; precision agriculture technologies; image analysis

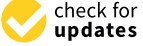



## 1. Introduction

In precision agriculture, an industry revolutionized by the digital age, the integration of advanced image processing and machine learning techniques, particularly convolutional neural networks (CNNs), is transforming crop management at the field level [1–3]. These deep learning models play a key role in automating agricultural tasks, such as crop health assessment, yield determination, and, most importantly, the harvesting process [4,5]. They are excellent at accurately identifying and localizing key crop characteristics, which are essential for efficient automated harvesting. This technological shift is particularly relevant in the context of tomato cultivation, where precise timing and manipulation during harvest are critical to ensure optimal yield and quality [6,7].

The agricultural industry faces significant challenges with the traditional, labor-intensive approach to harvesting, especially for delicate fruits, such as tomatoes. Manual harvesting is not only costly, but also fraught with human error and inefficiency. In addition, the global decline in available agricultural labor reinforces the need for more scalable and efficient harvesting methods. Robotic harvesting systems offer a viable solution to these challenges, promising to alleviate labor shortages and increase harvesting efficiency. However, the success of these systems depends on their ability to accurately and reliably identify and process crops. This requires the development of sophisticated algorithms capable of skillfully processing and analyzing images to accurately determine the ripeness and precise location of each tomato, ensuring efficient harvesting without damage.

## 1.1. Literature Review

In agricultural research, much of the literature emphasizes the importance of accurate fruit detection and localization [8–15]. Indira et al. [16], demonstrated the use of CNNs for fruit identification from visual data and emphasized the need for robust preprocessing to ensure consistency in image brightness and resolution. Preprocessing techniques, such as normalization and pixel-intensity adjustment, are crucial for providing a consistent baseline for subsequent computational analyses, which is reflected in a wide range of image analysis research. Naranjo et al. [17] emphasized the growing relevance of innovative technologies in the agri-food sector and highlighted artificial intelligence as a cornerstone of modern agricultural practices. The deep learning (DL) approach has the ability to infer complex patterns and representations from data, and it has been identified as a key driver of innovation, particularly in image-based applications. Rapado et al. [18] presented a novel approach to overcoming the limitations of traditional robotic vision systems in occluded agri-food environments using multiview perception and three-dimensional multi-object tracking. This method significantly improved object representation and localization accuracy, and achieved maximum errors of only 5.08% when estimating the number of tomatoes and up to 71.47% when tracking tomatoes in highly occluded environments. Afonso et al. [19] demonstrated the effectiveness of DL, particularly the MaskRCNN algorithm, in detecting and counting tomatoes in greenhouse images. Their results outperformed those obtained in previous studies in laboratory conditions or from higher-resolution images and demonstrated the ability of MaskRCNN to implicitly learn object depth to eliminate the background. Benavides et al. [20] presented an innovative computer-vision system designed to automate the detection and localization of fruits in a tomato crop in Mediterranean greenhouses. They designed the system to solve three main tasks: first, the detection of ripe tomatoes; second, the determination of their position in the XY plane of the image; and third, the localization of tomato-flower stalks for the same coordinates. To solve the first two problems, the authors used a full set of digital-image-processing tools, including enhancement, edge detection, segmentation, and detailed tomato characterization, which is very important for distinguishing the degree of ripeness and other visual factors under growing conditions. Li et al. [21] developed an improved YOLOv5s-tomato model that successfully detects the four maturity stages of greenhouse tomatoes with an accuracy of 95.58% and a mean accuracy (mAP) of 97.42%. This model, featuring improved Mosaic data and efficient IoU losses, outperforms the original YOLOv5s model in both accuracy and speed, making it suitable for precision-tomato-picking machine operations. Zheng et al. [22] developed a novel algorithm framework for automated fruit picking, focusing on greenhouse tomatoes, which combines a lightweight tomato-instance-segmentation model, YOLO-TomatoSeg, with an accurate tomato localization approach. This framework, integrating RAFT-Stereo disparity estimation and least-squares point cloud fitting, addresses the limitations of current neural-network-based fruit-recognition algorithms and traditional stereo matching methods, enhancing both segmentation accuracy and localization precision for tomato-picking robots. Zhao et al. [23] developed a robust tomato-recognition algorithm for autonomous harvesting robots, designed to operate in complex agricultural environments with disturbances like variable illumination and overlapping fruits. Utilizing a*-component and I-component feature images from Lab* and YIQ color spaces, fused through wavelet transformation and refined with an adaptive threshold algorithm, this method efficiently segments tomatoes from backgrounds, demonstrating a high recognition rate suitable for low-cost robotic harvesting. Wu et al. [24] introduced a novel automatic algorithm for the recognition of ripening tomatoes, designed to enhance robotic harvesting efficiency. This method employs a bi-layer classification strategy, incorporating color and textural features, feature analysis and selection, and a weighted relevance vector machine (RVM) classifier, effectively identifying tomato-containing regions and classifying them with high accuracy, paving the way for the automation of tomato detection and harvesting. Xiang et al. [25] developed an innovative recognition algorithm for tomato-plant stems, addressing the challenge of color similarity between stems and leaves. Utilizing a hybrid

joint neural network that combines the duality edge method with deep learning models, this algorithm effectively detects stems, reducing false negatives and positives, and is optimally suited to various automated processes in fruit and vegetable production, including targeted fertilization, pruning, and harvesting. Kanda et al. [26] introduced an intelligent deep-learning-based method for the recognition of nine common tomato diseases, utilizing a residual neural network algorithm. This research, focusing on various levels of network depth and learning rates, demonstrates high accuracy in disease identification, outperforming many previous methods, although it faces challenges in distinguishing between closely related disease classes. Zu et al. [27] developed a method using the Mask R-CNN algorithm for detecting and segmenting mature green tomatoes, which are challenging to identify due to their color similarity with branches and leaves, as well as their often obscured and overlapping positions. Utilizing a mobile robot for diverse image collection in a greenhouse, the method employs RestNet50-FPN as its backbone network and achieves efficient detection and segmentation, demonstrating its potential for direct application in automated tomato-harvesting robots. Li et al. [28] proposed an enhanced YOLOv5s-based algorithm, YOLOv5s-CQE, for the recognition and localization of targets in intelligent tomato picking. This algorithm incorporates the CARAFE module to optimize upsampling, along with EIoU and Quality Focal Loss, to address uneven sample distributions, resulting in improved accuracy and training speed, demonstrating high precision, fast detection, and robustness, thereby offering a solid foundation for the development of intelligent tomato-picking systems.

### 1.2. Objectives

The aim of this research is to develop advanced tomato-image-preprocessing techniques in the context of gravimetric analysis applied to robotic harvesting. The study is unique as it aims to comprehensively analyze the impact of image-preprocessing techniques on the performance of two-dimensional (2D) correlation matrices, especially in CNNs, which has not been previously addressed in such a context. We propose new techniques and hypotheses that significantly improve the accuracy and efficiency of automated tomato harvesting.

1. The development of improved preprocessing techniques: This research introduces an improved tomato-image-preprocessing technique for robotic harvesting systems, involving the integration of part affinity fields (PAFs) and confidence maps. This provides improved accuracy for tomato-center-of-gravity localization, extending the capabilities of previous approaches focused mainly on fruit detection and localization.
2. Quantitative improvement and practical applicability: In our study, a series of structured experiments were conducted to evaluate the improvements made by these methods. We evaluate the practical applicability of these methods in different real-world agricultural scenarios, providing a detailed understanding of the relationship between pretreatment and CNN performance.

Our study aims to address the following gaps:

3. The lack of attention to preprocessing in CNNs: Our study fills this gap by focusing on how image preprocessing affects the accuracy of CNN models, especially in precision agriculture.
4. Improved robustness and adaptability: The data augmentation and multispectral image-enhancement techniques used in this study improve the robustness and adaptability of the proposed model, given the variability in environmental conditions and tomato varieties.
5. Innovative approach to key point localization: This study focuses on the localization of key points, particularly the center of gravity, offering a new perspective and achieving high accuracy in this area.

The initial hypotheses of this study suggest that advanced image-preprocessing techniques integrated with CNNs can significantly improve the accuracy of tomato detection

and localization, improving the efficiency of automated harvesting in real-world environments. This research represents a significant advancement in agricultural automation, affecting crop management and harvesting. It lays the foundation for future advances in agricultural technology.

Thus, the aim of the study is not only to present a new image-preprocessing technique, but also to demonstrate its effectiveness in practical applications, filling a critical gap in agricultural automation.

## 2. Materials and Methods

### 2.1. Image Acquisition and Processing

In this study, a unique database containing 2260 tomato images was developed. Of these, 700 images were taken in greenhouses in Niigata (Niigata Prefecture, Japan). The remaining 1560 images were carefully selected from various publicly available sources, including Kaggle and Github [29–31], to ensure the greatest representativeness and diversity. During the selection, special attention was paid to aspects such as tomato ripeness, lighting conditions, shooting angles, and background features, as demonstrated in Figure 1. As a result, the database covers images of tomatoes grown under a variety of conditions, including greenhouse conditions. This approach to database compilation was deliberately chosen to increase the reliability and universality of the analysis results.

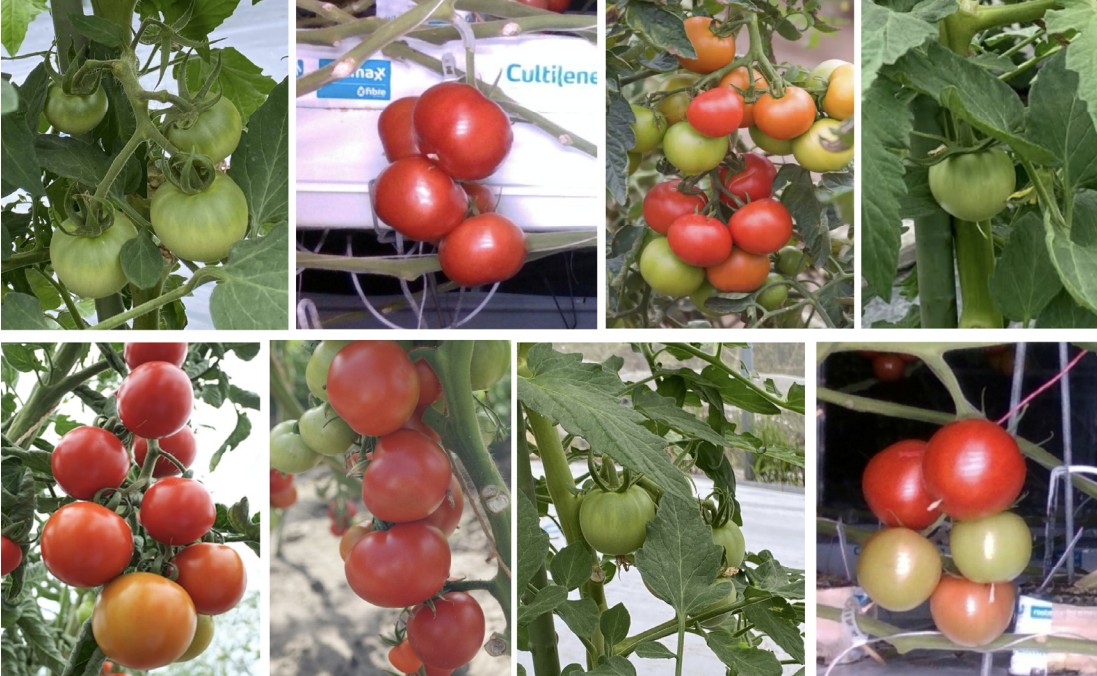

**Figure 1.** Examples of dataset images.

The data-labeling process involved both automatic and manual methods, with initial labeling performed by a pre-trained CNN model and subsequent manual validation by agricultural experts. We established clear inclusion criteria, focusing on healthy, mature tomatoes under different light conditions, and excluded any images that were blurred or contained multiple overlapping fruits. This approach ensures the relevance and applicability of our dataset to real-world agricultural scenarios, addressing the limitations of existing datasets in representing the diverse conditions encountered in tomato harvesting.

The images were standardized and underwent a number of preprocessing steps to ensure the uniformity of resolution and brightness. To preserve quality, each image was resized to $224 \times 224$ pixels using modern interpolation techniques. The pixel intensities

were normalized to a maximum value of 255. These processes were implemented in Python using OpenCV and scikit-image libraries.

The methodology developed in this study was based on a CNN, which was chosen for its effectiveness in pattern recognition tasks. The CNN consisted of several convolutional layers with rectified linear unit (ReLU) feature activation, max-pooling layers and fully connected layers for feature extraction, dimensionality reduction, and classification, respectively.

The dataset was divided into training (80%), validation (10%), and test (10%) subsets. Training was performed in batches and after each epoch, performance on the validation set was evaluated to fine-tune the model and prevent overfitting.

A personal computer (Intel i7, 3.60 GHz, 32 GB DDR4, GeForce RTX 2080) running the Windows 11 operating system was used as the computing machine. The Python programming language was used to build the system.

The reproductive structure of a tomato is illustrated in Figure 2. It consists of several biologically distinct components that link the main stem to the fruit. These components include the peduncle, abscission region, and sepal. The branch emanating from the main stem bears one or more pedicels, from each of which a separate fruit develops [32]. Anatomically, the pedicel is divided into distal and proximal parts relative to the abscission zone. This zone is where the physiological detachment of the mature fruit occurs through the abscission process [33].

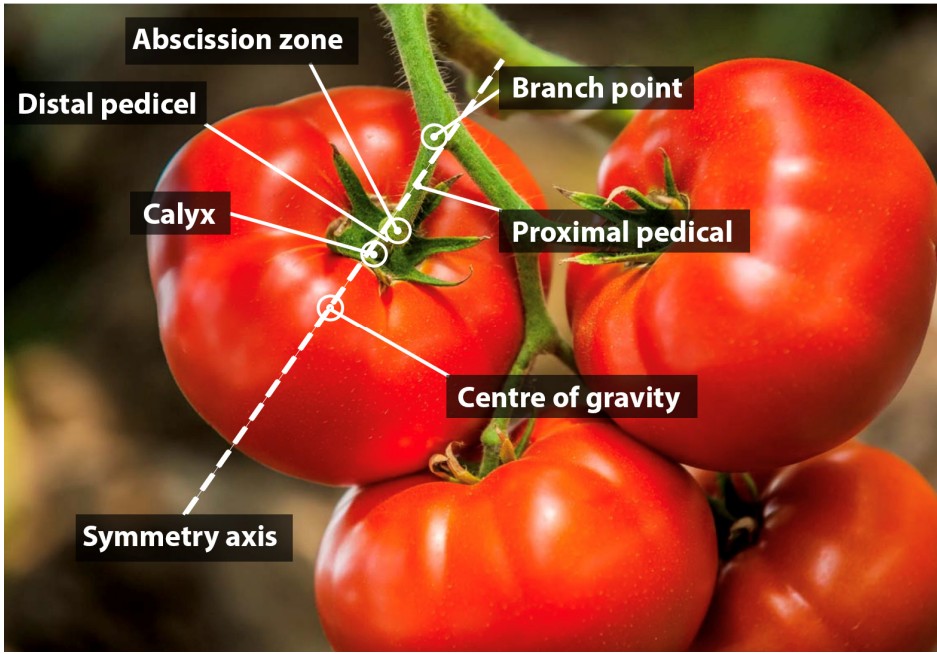

**Figure 2.** Structure of the tomato-fruiting system and main compositions used to identify key points.

In the context of developing robotic harvesting systems, determining the center of gravity of an object is a key task. This information is critical for ensuring the balance and stability of the robot, particularly when manipulators or rakes are used to lift and move the crop. Accurate information about the location of the center of gravity allows control and manipulation to be optimized, which minimizes the risk of crop damage and improves harvesting accuracy. Additionally, this information contributes to energy efficiency by allowing the robot to move more economically without wasting unnecessary energy on maintaining balance or adjusting its position.

Determining the center of gravity is also important for adapting robotic systems to various types of crops, which can vary significantly in size, shape, and weight. This increases the flexibility of the system and allows it to adapt to a variety of harvesting conditions. Additionally, the correct understanding of this aspect prevents potential accidents and

mishaps, which ensures the safety of both the robot and the environment. Considering the center of gravity in the initial design stages contributes to improving the efficiency and reliability of robotic harvesting systems, which increases the overall productivity and safety of agricultural operations.

### 2.2. Enhanced Pose Estimation via CNNs

The proposed methodology uses dual 2D matrices to improve estimation of tomato position using both confidence maps and PAFs, which builds on the foundations established in [34,35]. Confidence maps are probabilistic heat maps that locate key points that define the point in the center of gravity, with each tomato in the dataset represented by a unique map that consists of multiple channels [36]. These channels directly correlate with various key points on the tomato. The spatial distribution of these key points is mapped to the individual channels for each tomato image, with the true value of $S^*$ for each pixel given by

$$S_{j,k}^*(p) = \exp\left(-\frac{\left|p - x_{j,k}\right|_2^2}{\sigma^2}\right),$$ (1)

where $j$ and $k$ represent the key point type and tomato index, respectively, and $\sigma^2$ is an adjustable parameter that determines the confidence spread for the key point.

Furthermore, CNNs are used to optimize the training and evaluation of these confidence maps, and provide high accuracy and robustness [37]. The collective confidence map for a given key point across all tomato plants is computed using the maximum operation:

$$S_j^*(p) = \max_k S_{j,k}^*(p).$$ (2)

The physical orientation of each tomato is determined by the key points detected by a PAF. The PAFs represent the relationships between key points as 2D vectors that indicate the location and orientation of the tomato parts [38]. The number of links each relationship has is $C$, with the PAF $L^*$ for each relationship defined as

$$L_{c,k}^*(p) = \begin{cases} v & \text{if p on limb } c, k \\ 0 & \text{otherwise,} \end{cases}$$ (3)

where $v$ is a unit vector directed from one key point to the other:

$$v = \frac{\left(x_{j2,k} - x_{j1,k}\right)}{\left|x_{j2,k} - x_{j1,k}\right|_2}.$$ (4)

The criterion of a point that belongs to a pattern is defined by its perpendicular distance to the line segment between the key points, which must not exceed $\sigma_1$:

$$0 \le v \cdot \left(p - x_{j1,k}\right) \le l_{c,k} \quad \text{and} \quad \left|v_\perp \cdot \left(p - x_{j1,k}\right)\right| \le \sigma_l.$$ (5)

To determine the true value $L^*$, the average PAF over all images for a particular part is calculated:

$$L_c^* = \frac{1}{n_c(p)}\sum_k L_{c,k}^*(p),$$ (6)

where $n_c(p)$ corresponds to the number of non-zero PAF values at point $p$ for a given part in all images.

The integration of CNNs not only improves the accuracy of key point detection but also greatly improves the PAF calculation, which is very important for accurate robotic tomato harvesting based on gravimetric analysis.

### 2.3. Comparative Analysis of Preprocessing Techniques in Image-Feature Delineation

In the realm of image processing, preprocessing plays a pivotal role in enhancing the quality of an image or preparing an image for further operations. In this study, three pre-eminent image-processing methods were meticulously chosen and their efficacy in delineating features was determined (Figure 3).

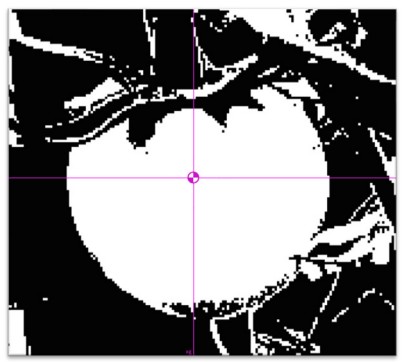 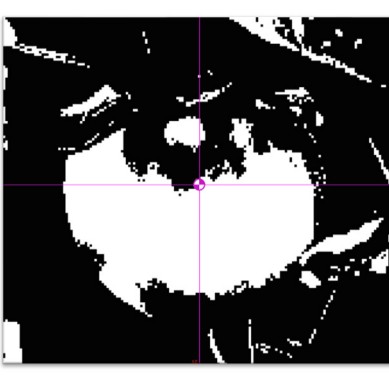 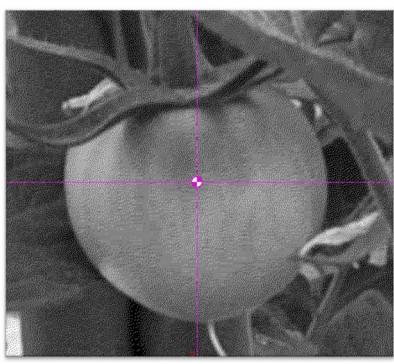

Threshold: "110"  Threshold: "120"  Dithering

**This results in an input image with the centroid marked (marked with a purple circle):**

**Figure 3.** Assessment of preprocessing methods for feature extraction in image analysis.

1.  Thresholding at an intensity of 110: This technique involves setting a threshold at an intensity value of 110. Pixels with an intensity that exceeds this value are labeled as the object (in this case, a tomato), whereas those with an intensity below this value are designated as the background. This approach is particularly advantageous for discerning the primary contours of the object in scenarios in which specific lighting conditions prevail.
2.  Thresholding at an intensity of 120: The functionally of this technique is similar to that mentioned above. The technique uses a slightly elevated threshold value. It has been proven to be particularly effective for images that have pronounced contrast or were captured under intense illumination.
3.  Dithering: Dithering is a sophisticated method that capitalizes on quantization errors to metamorphose the image into a monochromatic version, albeit with the enhanced retention of intricate details and textures. This method is instrumental in enabling the analytical model to pinpoint and identify complex regions or areas of the tomato that exhibit low contrast, which might otherwise be overlooked when using rudimentary binarization techniques.

It is crucial to understand that the efficacy of each method is contingent on the specific attributes of the image in question and the overarching objective of the image-processing task.

### 2.4. Architecture for Tomato Pose Detection

In this study, a modified methodology is developed to improve the efficiency of tomato gravimetric analysis (Figure 4). The architecture includes a backbone for feature extraction, an initial prediction stage to construct confidence maps and PAFs, and a series of refinement stages to improve accuracy. In the initial stage, feature sets are extracted in the backbone, and trust maps and PAFs are predicted. In subsequent stages, these predictions are refined using previous feature maps and results as inputs [39,40].

The network-training process uses a loss function that carefully compares the predicted confidence and PAF values with the corresponding "true" values of $S_j^*$ and $L_c^*$, as defined in the loss function:

$$f_s^t = \sum_{j=1}^{J} \sum_p W(p) \big| S_j^t(p) - S_j^*(p) \big|_2^2$$
$$f_L^t = \sum_{c=1}^{C} \sum_p W(p) \big| L_c^t(p) - L_c^*(p) \big|_2^2 \qquad (7)$$
$$f = \sum_{t=1}^{T} f_s^t + f_L^t,$$

where $t$ denotes the refinement stage, $S_j^t(p)$ and $L_c^t(p)$ are the confidence maps and PAF predictions at each stage, respectively, and $W(p)$ is a binary mask used to prevent penalties for true positives in unannotated regions. The multi-tomato pose-recognition process uses these confidence maps and PAFs, and the predictions of the last step are used to output the fully trained model. This process is discussed in more detail in Section 3.

This study methodically assesses the individual contribution of each component of our hybrid CNN-PAF model. By systematically removing or modifying certain elements, CNN layers, PAFs, and preprocessing methods, we evaluate their impact on model performance. This analysis provides valuable insights into the importance of each component and further justifies our design choices.

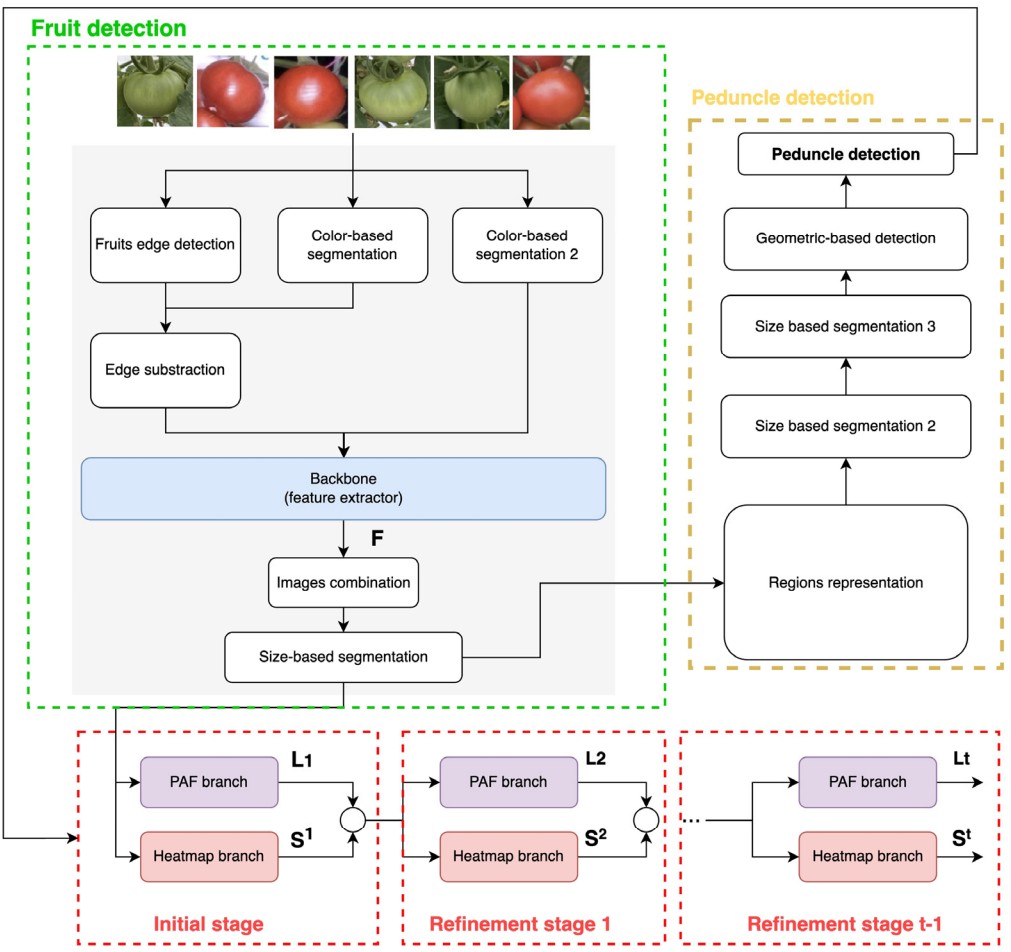

**Figure 4.** Methodology diagram.

Clarification stages (Figure 4):

1. Initial stage: This stage contains a PAF branch and heat-map branch, denoted by L1 and S1, respectively. A PAF is used to encode parts of the objects and their connections, whereas the heat map branch is used to localize key points.
2. Refinement stage 1: As with the initial stage, this stage involves reusing PAF branches and heat maps (L2 and S2) to further refine the results.

3.   Refinement stage *t*+1: This represents the final stage in an iterative refinement process, where '*t*' indicates the number of intermediate refinement stages. Each subsequent stage involves the further improvement of localization accuracy.

Figure 4 illustrates a complex and multi-layered approach to detecting and localizing fruits and their peduncles, which involves a series of image-processing steps and feature analyses.

## 3. Results

Results were obtained for the developed hybrid model that combined CNNs and dual 2D matrices for the classification and PAFs. The results of the visualization of the tomato detection and positioning method within the DL and the image processing for subsequent robotic harvesting systems are presented in Figure 5. The figure shows unripe tomatoes with color-coded labels that indicate key analysis points: the fruit centers (1: green dot), sepals (2: red dot), and abscission zones (3: sky blue dot), and the branching points of the branches (4: black dot). These labels help to identify and evaluate the plant structure for accurate robotic harvesting. The supporting images (a, b, c, and d) show various image-processing and -analysis steps, such as segmentation and contour detection, with the identification of the points in the center of attachment. The results suggest the high accuracy of object recognition achieved by the model, which indicates its application potential in real agricultural conditions. Errors in key point detection were more common in the images with small or clustered fruits; however, overall, the model demonstrated a high level of performance and provided useful data for robotic control.

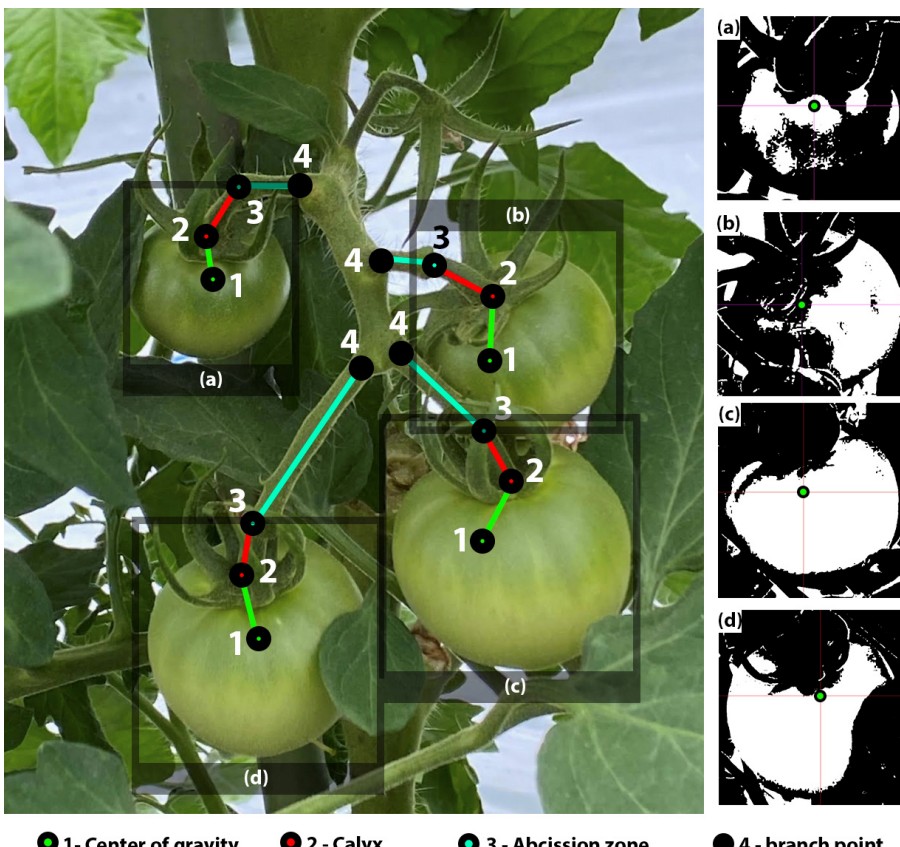

● 1- Center of gravity   ● 2 - Calyx   ● 3 - Abcission zone   ● 4 - branch point

**Figure 5.** Results of deep learning and the evaluation of the effect of jointly using the branching point to estimate the tomato pose and center-of-gravity point, (**a**–**d**): Evaluation of threshold preprocessing method for feature extraction in image analysis.

Figure 6 shows six tomato-brush images that demonstrate various stages of tomato ripening and arrangement, not including indistinguishable tomato key points. In each image, the tomato plants are marked using colored dots and connected by lines to highlight key anatomical points and their relationships. The lines that connect these points are represented in their respective colors, which indicate the anatomical relationships between these elements.

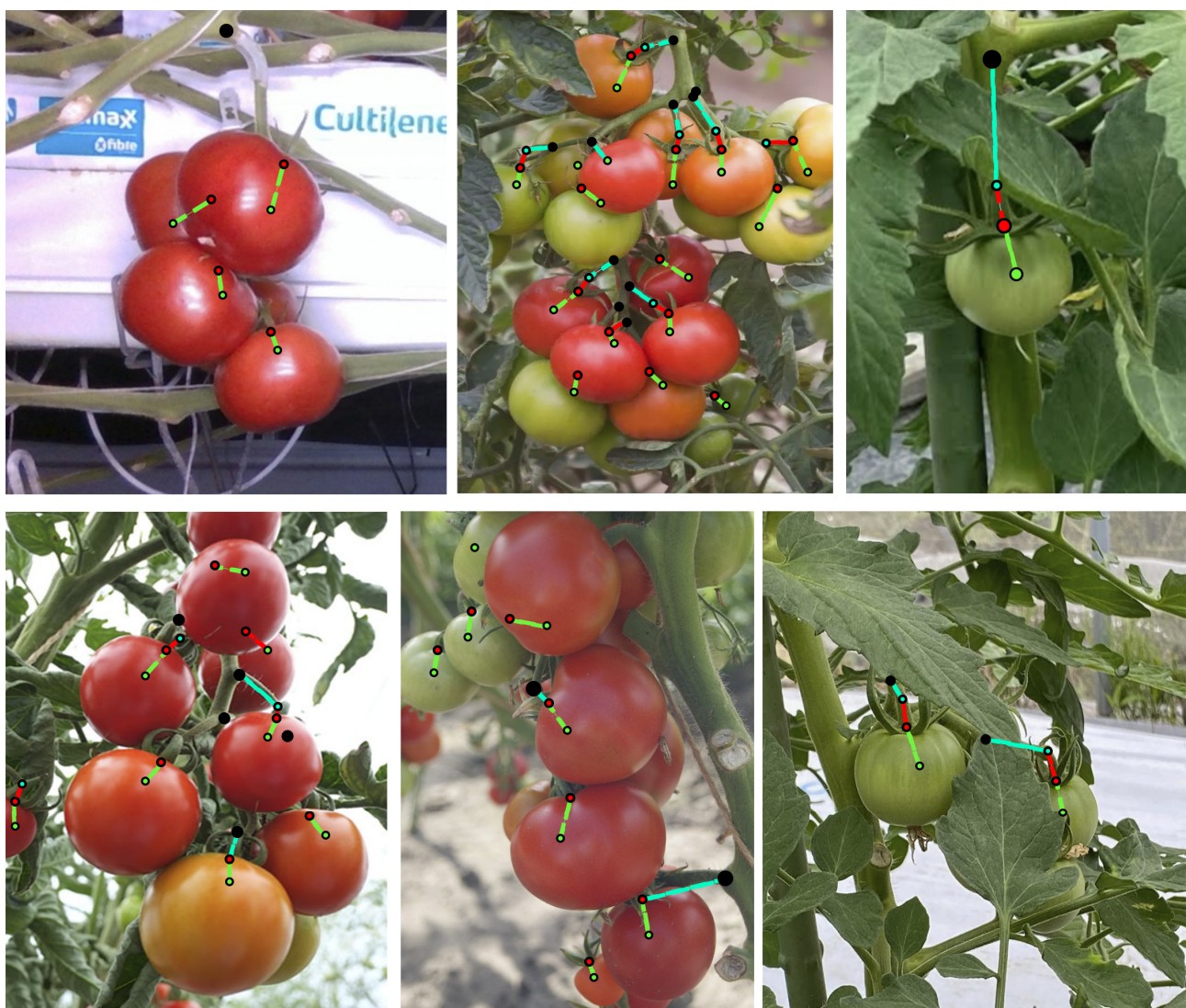

**Figure 6.** Results of recognizing tomato samples that included indistinguishable key points (parts).

Figure 6 illustrates the methods used to identify the positions of the tomatoes and their anatomical parts, which has significant applications for the development of robotic harvesting algorithms. The images show both successful and difficult feature recognition caused by overlapping or visually merging plant parts, which highlights the importance of using an integrated approach to visual information recognition and processing for the automation of agricultural processes.

In our study, the accuracy of the key point localization and the part-detection errors were quantified for the proposed method. The key-point-localization errors are presented in Table 1, where they are categorized into types: center-of-gravity point, calyx, abscission zone, and branching point. The errors were measured at the pixel level and ranged from 5.4–10.6 pixels horizontally and 5.8–13.1 pixels vertically. On average, the errors for all the key points were 5.3 pixels on the *x*-axis and 6.2 pixels on the *y*-axis. The smallest error

in pixels, approximately 2.6 pixels, was observed for the calyx, and the largest error was observed for the branching point (10.6–13.1 pixels), which was three times larger than the other key points. This high error rate at the branching point occurred due to its low visibility, often resulting from its merging with stem parts in adjacent areas. Despite this, the calyx areas and the tomato's center of gravity point had the lowest pixel errors, which highlights their geometric importance in enabling robotic systems to accurately identify harvest points.

**Table 1.** Position errors for the detected key points.

| Axis | Center of Gravity | Calyx | Abscission Zone | Branch Point | Total |
|---|---|---|---|---|---|
| *X* (pixel) | 5.3 | 2.6 | 4.1 | 10.6 | 5.65 |
| *Y* (pixel) | 6.2 | 3.0 | 4.7 | 13.1 | 6.75 |
| Overall average score (%) | 97.2 | 96.7 | 95.5 | 96.2 | 96.4 |

The last row in Table 1 presents the evaluation of the key point detection for a normalized distance. For this metric, the key point is considered to have been accurately detected if the error is less than half the length of the bounding box diagonal. An error in this context means the Euclidean distance between the detected point and the real point. The overall average score for all the key points was 96.4%, with the highest detection accuracy and potential for practical application achieved for the center-of-gravity point and abscess.

In this study, an innovative model was developed and tested for the advanced image preprocessing of tomatoes, with the aim of determining their center of gravity for robotic harvesting. The integration of a hybrid model that combined CNNs with PAFs was key to achieving high accuracy and adaptability under various agricultural conditions. This study underscores the importance of integrating multispectral analysis and data-enhancement techniques, which enabled the model to effectively adapt to a diverse range of environmental conditions and tomato varieties. The model's impressive recognition accuracy of 96.4% on an extensive dataset of 2260 images indicates its potential in real-world agricultural applications and may lead to significant reductions in manual labor and increased harvesting efficiency. This technology not only furthers the automation of agricultural processes but also paves the way for more research in the fields of gravimetric analysis and robotic harvesting. In conclusion, this study demonstrates considerable progress in the development of technologies for agricultural automation, which suggests a trajectory for future improvements that will have a substantial influence on the agricultural industry as a whole.

## 4. Discussion

In this section, we analyze and discuss the findings of our study, in which we focused on improving automated tomato harvesting through a novel hybrid model integrating CNNs and PAFs.

1.  The core achievement of our research is the development of a hybrid model that synergizes CNNs with PAFs, which resulted in a remarkable recognition accuracy on a diverse dataset. This high accuracy, which was consistent under various environmental conditions and tomato varieties, is evidence of the model's robustness and adaptability. The integration of DL with state-of-the-art image-processing techniques made our algorithm significantly outperform existing algorithms, which marks a substantial advancement in the field of agricultural automation.
2.  Despite the overall success of our model, it encountered challenges in key point detection, particularly for images that contained small or clustered fruits. These errors, detailed in Table 1, provide crucial insights into the limitations of the current model. The higher error rates observed for the branching points caused by their frequent merging with the stem parts highlighted areas for future refinement. By contrast,

the low number of pixel errors for the calyx and the tomato's center-of-gravity point indicate their geometric importance in robotic systems in terms of achieving accurate harvest point identification.

3.  The findings have significant implications for the field of robotic harvesting. The model's ability to accurately identify and position tomatoes paves the way for more efficient, precise, and scalable robotic harvesting solutions. This advancement could lead to a reduction in manual labor and an increase in harvesting efficiency, which are crucial for meeting the growing demands of the agricultural industry.

4.  Another critical aspect of our study is the model's performance under various real-world agricultural conditions. The integration of multispectral analysis and data-enhancement techniques enabled the model to effectively adapt to various environmental conditions and characteristics of tomato varieties. This adaptability is essential for the practical application of the technology in diverse agricultural settings.

5.  Our study marks a significant step forward in agricultural automation and also provides opportunities for further research. Areas for future improvement include enhancing the model's ability to process images that contain small or clustered fruits and refining the detection of more challenging key points, such as branching points. Additionally, extending the model's application to other crops and agricultural practices could broaden its impact on the industry.

To summarize, our study demonstrates a significant breakthrough in the automation of agricultural processes, particularly in the context of tomato harvesting. The developed model not only offers high accuracy and adaptability but also provides a foundation for further advancements in the field. This model has the potential to transform agricultural practices by contributing to the sustainability and efficiency of food production globally.

## 5. Conclusions

This research represents a significant leap forward in the field of agricultural automation, particularly in the context of tomato harvesting. Our study centered on the development and extensive testing of an innovative hybrid model that integrates CNNs with PAFs, which is a combination that has been proven to be highly effective in accurately determining the center of gravity of tomatoes for robotic harvesting.

In this study, we placed a particular emphasis on the integration of multispectral analysis and data-enhancement techniques, which allowed the model to effectively adapt to a variety of environmental conditions and characteristics of various tomato varieties. As a result, the model achieved an impressive recognition accuracy, reaching 96.4% on an extensive dataset of 2260 images. Despite the successes of the study, we identified areas for improvement, particularly in enhancing the model's ability to handle challenging scenarios, such as small or clustered fruits. Future research could focus on refining key point detection and extending the model's applicability to other crops and agricultural processes.

In conclusion, our study contributes significantly to the advancement of technologies in agricultural automation. The high accuracy, efficiency, and adaptability of the developed model not only makes more effective and sustainable agricultural practices possible, but also opens up new opportunities for research and development in this field. The implications of this study extend beyond tomato harvesting, which suggests a promising path for the broader application of such technologies in the agricultural industry, which could transform the way that we approach food production and resource management on a global scale.

**Author Contributions:** Conceptualization, N.B.; methodology, N.B. and H.H.; software, N.B.; verification, H.H.; formal analysis, N.B. and H.H.; investigation—H.H.; data curation, N.B.; writing—preparation of original draft, N.B.; writing—review and editing, N.B. and H.H.; visualization, N.B.; funding acquisition H.H. All authors have read and agreed to the published version of the manuscript.

**Funding:** This research received no external funding.

**Institutional Review Board Statement:** Not applicable.

**Informed Consent Statement:** Not applicable.

**Data Availability Statement:** The data presented in this study are publicly available: https://www.kaggle.com/datasets/aanxiaanxi/tomato-image-data-set-used-for-robotic-harvesting (accessed on 21 December 2023).

**Acknowledgments:** We express our sincere gratitude to Yaroslav Zakharevich, a first-year doctoral student at the Graduate School of Science and Technology, Niigata University, for his invaluable contributions in the early stages of our study. We also thank Masakazu Imaguchi, director of IMA Co., whose support and help in funding this article were very important. Their collective contributions greatly enriched our work and enabled the successful completion of this study.

**Conflicts of Interest:** The authors declare no conflicts of interest.

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
