# Peer review of "Advanced Preprocessing Technique for Tomato Imagery in Gravimetric Analysis Applied to Robotic Harvesting"

_applsci, doi:10.3390/app14020511_

Round 1

Reviewer 1 Report

Comments and Suggestions for Authors

This is an well-written article that proposes a hybrid CNN model with part affinity fields for accurately determining the center of gravity location in tomatoes to enable more efficient robotic harvesting systems. However, there are major issues that must be addressed before the paper can be considered for publication into a journal:

1. There is a complete lack of background literature. A section is required to discuss existing studies with the aim to highlight the gap. Without this, the author's claim of novelty is very weak. They need to show through literature how their approach is different. 

2. There is a lack of comparison with existing similar studies. It is not clear what their contribution(s) is/are and this is in part due to lack of background literature related to the field. The authors should establish the state of the art and consider this as the baseline for measuring their method. Without this, their claim of 'improved' performance is invalid.

3. The authors claimed in the introduction that their method was tested on several datasets but only one was presented. This means that the conclusion does not match the experiments presented. Consider conducting the experiments with an existing dataset that has been utilised by other researchers so that you can compare your results. If authors must use their own dataset, a strong case must be made about why the existing datasets are not directly applicable.

4. The data is extremely small for cnn research. The data acquisition process lacks details. How was the data labelled? Where did they get the data from (I.e., specific details of the sources and how many they took from each repo). What is the inclusion and exclusion criteria.

Comments on the Quality of English Language

moderate editing of English language required but this does not affect readability. 

Reviewer 2 Report

Comments and Suggestions for Authors

The paper introduces a hybrid model combining CNNs with part affinity fields (PAFs) for improved tomato harvesting. While the integration of deep learning with image processing is not entirely novel, the specific application to tomato harvesting, the emphasis on multispectral analysis, and the detailed exploration of preprocessing techniques contribute to the paper's originality. The utilization of PAFs for tomato pose estimation also adds a novel dimension to the research. 
The paper is well-organized, providing a comprehensive overview of related work, methodology, and results. The inclusion of figures aids in understanding the proposed model and its components.
The technical contribution is substantial, particularly in the development of a hybrid model that significantly improves tomato detection and positioning. The integration of multispectral analysis and data enhancement techniques showcases a thoughtful approach to adaptability under various agricultural conditions. The detailed comparative analysis of preprocessing techniques adds value to the technical contribution.

However, the paper falls short in not conducting an ablation study, which is a significant omission as it would provide insights into the contribution of each component and justify design choices.

The paper makes a valuable contribution to the field of agricultural automation, specifically in tomato harvesting. However, it is recommended that the authors conduct an ablation study to analyze the impact of individual components, such as PAFs and multispectral analysis, on the overall performance. This would enhance the credibility of the proposed model and provide a more in-depth understanding of its strengths and limitations. Additionally, a clearer emphasis on the paper's unique contributions in the context of existing literature would strengthen the submission.

Overall, with the suggested improvements, the paper has the potential to be a significant and impactful contribution to the field of agricultural automation.

Additional Specific Comments:

1. What is the main question addressed by the research?

The paper introduces a hybrid model combining CNNs with part affinity fields (PAFs) for improved tomato harvesting.

    2. Do you consider the topic original or relevant in the field? Does it
    address a specific gap in the field?

The technical contribution is substantial, particularly in the development of a hybrid model that significantly improves tomato detection and positioning. The integration of multispectral analysis and data enhancement techniques showcases a thoughtful approach to adaptability under various agricultural conditions. The detailed comparative analysis of preprocessing techniques adds value to the technical contribution.

    3. What does it add to the subject area compared with other published
    material?

While the integration of deep learning with image processing is not entirely novel, the specific application to tomato harvesting, the emphasis on multispectral analysis, and the detailed exploration of preprocessing techniques contribute to the paper's originality. The utilization of PAFs for tomato pose estimation also adds a novel dimension to the research.

    4. What specific improvements should the authors consider regarding the
    methodology? What further controls should be considered?

the paper falls short in not conducting an ablation study, which is a significant omission as it would provide insights into the contribution of each component and justify design choices.

    5. Are the conclusions consistent with the evidence and arguments presented
    and do they address the main question posed?

Yes

    6. Are the references appropriate?

Yes

    7. Please include any additional comments on the tables and figures.

The paper is well-organized, providing a comprehensive overview of related work, methodology, and results. The inclusion of figures aids in understanding the proposed model and its components.

Round 2

Reviewer 1 Report

Comments and Suggestions for Authors

The authors have made reasonable adjustments to address the main concerns to some extent as indicated below.

- The new section on `literature' reviewed relevant studies. The authors claimed to have compared their results with the existing studies but this was not done appropriately i.e., comparison with existing methods will require replication of those methods on the custom dataset used for this study. While I understand the difficulty in replicating these experiments, the authors should at least indicate that such comparison was not feasible (perhaps due to time limitation or other factors). 

Recommendation: remove all mention of comparison with existing studies and replace them with a clear statement to show why this was not possible/feasible.

- Effort has been made to clarify the data collection process (although not explicitly) to allow for reproducibility. 

Recommendation: In the interest of open science, it is recommended that the authors publish the custom dataset to allow for reproducibility of results and also contribute to advancement of research in this area. I note that their research would be impossible if it was not for other publiclly available datasets. The authors should consider publishing their custom dataset in repositories such as Github, Mendeley data, Kaggle etc.

Generally, the paper is well written and will benefit from the recommended `minor' changes before publication.

Comments on the Quality of English Language

Some Minor editing of English language required especially in the newly added sections.
